# The Implementation and Impact of a Revised National Childhood Immunization Schedule in an Urban Asian Community

**DOI:** 10.3390/vaccines10071148

**Published:** 2022-07-19

**Authors:** Ngiap Chuan Tan, Qifan Tan, Wai Keong Aau, Chung Wai Mark Ng

**Affiliations:** 1SingHealth Polyclinics, Singapore 150167, Singapore; tan.qifan@singhealth.com.sg (Q.T.); aau.wai.keong@singhealth.com.sg (W.K.A.); ng.chung.wai@singhealth.com.sg (C.W.M.N.); 2SingHealth Duke-NUS Family Medicine Academic Clinical Programme, Singapore 150167, Singapore

**Keywords:** vaccine, immunization schedule, childhood vaccination

## Abstract

Changes to the national childhood immunization schedule (NCIS) can have a potential impact on vaccine uptake in the community. The NCIS in Singapore has undergone several revisions over the years, with the most recent modification on 1 November 2020. The new NCIS includes, as routine, the influenza and the varicella vaccine, as well as two combination vaccines, the measles, mumps, rubella and varicella vaccine (MMRV), and the hexavalent diphtheria, acellular pertussis, tetanus, haemophilus influenza b, injectable polio, and hepatitis B vaccine (6-in-1). This retrospective database study aims to assess the effect of the new NCIS on (a) the vaccination uptake of children at 6 and 12 months and (b) the cost difference to the healthcare system and to parents. One-year vaccination data from two cohorts of children immunized according to the old (*n* = 10,916) and new NCIS (*n* = 10,299) were extracted, respectively, from their electronic medical records. The vaccine uptake at 6 and 12 months increased by 10.8 and 2.1%, respectively, with the new NCIS as compared to the old NCIS. The mean number of required visits to the primary care clinic for each child was reduced from six to four. There is an estimated 6.41% cost reduction with the new NCIS.

## 1. Introduction

In Singapore, children are vaccinated according to the national childhood immunization schedule (NCIS) [1]. Over the years, the NCIS has undergone several revisions [2,3]. Prior to the November 2020 revision, the NCIS covered vaccinations against hepatitis B (HepB), diphtheria, tetanus, acellular pertussis, injectable polio, haemophilus influenza b (5-in-1), measles, mumps, rubella (MMR) and pneumococcal (PCV13) disease [3]. The most recent NCIS revision was in November 2020. In this revision, several new vaccines were included as part of routine childhood vaccinations. These included the influenza and varicella vaccines, as well as two combination vaccines: viz the quadrivalent measles, mumps, rubella, and varicella vaccine (MMRV) and the hexavalent diphtheria, acellular pertussis, tetanus, haemophilus influenza b, injectable polio, and hepatitis B vaccine (6-in-1) [3]. The additions to the new NCIS are important, as influenza poses a considerable healthcare burden in Singapore [4] which results in significant morbidity [5], and a significant proportion of preschool children remain susceptible to varicella [6]. The introduction of the varicella vaccine into the national childhood vaccination schedule is intended to mitigate the risk outbreaks which can occur with inadequate herd immunity, even in developed countries [7]. The new NCIS also aims to align clinic visits with specific childhood developmental assessment touchpoints and introduce new vaccines without increasing the required number of visits.

While the influenza, varicella, MMRV and the 6-in-1 vaccines were not routine vaccines in the old NCIS, all of these vaccines (except MMRV) were already available in polyclinics on a non-subsidized basis. In addition, PCV13 was not subsidized prior to November 2020, even though it was part of routine vaccination then. In other words, there was no subsidy for influenza, PCV13, varicella, or the 6-in-1 vaccine with the old NCIS.

Parents can pay for PCV13 and the 6-in-1 vaccines using their Medisave account, if there are sufficient funds within. Medisave is a compulsory national savings scheme where individuals with an income are required to set aside a portion of their earnings which may be used for specific approved healthcare expenditure [8]. With the new NCIS, all included vaccines are fully subsidized and parents no longer need to pay via their Medisave accounts [9].

Overall, the new NCIS has been implemented based on the postulation that it will increase the rate of childhood vaccination uptake, reduce the frequency of healthcare visits [10], expand the number of administered vaccines, and ultimately reduce pediatric morbidity and mortality from preventable infections. The use of combination vaccines has been shown to improve the timeliness of vaccinations [11].

This study aims to compare the uptake of various childhood vaccines and the frequency of visits to primary healthcare providers for vaccine administration based on a one-year period before and after the new NCIS was introduced [12]. The secondary aims are to estimate the difference in the expenditure on vaccine administration after the commencement of the NCIS and to identify the demographic factors associated with vaccine uptake [13]. The results will provide insight into the effect on vaccine uptake and guide the development of future public health policy, media publicity and health outcomes resulting from an updated national childhood immunization program [14].

## 2. Materials and Methods

### 2.1. Study Sites and Population

This database study is a retrospective review of the electronic medical records (EMRs) of completed childhood vaccinations in SingHealth Polyclinics (SHPs). At the time of the study, SHPs were a network of eight public primary care clinics (polyclinics) in the central and eastern regions of the densely populated urbanized island state. The eight polyclinics served the healthcare needs of 351,624 multi-ethnic Asian children aged 0 to 3 years old from November 2019 to December 2021.

The study population comprised children of Chinese, Malay, Indian, and other minority ethnic groups who received their childhood vaccinations at the eight polyclinics during the stipulated study period. Those born between 1 November 2019 and 31 October 2020 received their vaccinations according to the old NCIS. Records of their vaccinations were retrieved from 1 November 2019 to 29 December 2020. The children’s ages were determined at the end of the 12-month study period: 29 December 2020 for the old NCIS and 29 December 2021 for the new NCIS. Under the new NCIS, the study population comprised children born between 1 November 2020 and 31 October 2021. Records of their vaccinations were extracted from 1 November 2020 to 29 December 2021. The data extractions from EMRs were extended for both cohorts to include those with delayed vaccinations due to various reasons, such as acute illnesses.

### 2.2. Vaccination Schedule

The new NCIS was rolled out on 1 November 2020 [15]. The new NCIS includes the influenza and varicella vaccines; combined diphtheria, tetanus, pertussis, poliovirus, haemophilus b and hepatitis B (6-in-1) vaccine; and combined measles, mumps, rubella and varicella vaccine (MMRV). The vaccination schedules for the former and current NCIS are shown in Table 1.

### 2.3. Data Extraction, Processing and Audit

In SHPs, clinical data such as consultation notes, including vaccinations, are recorded in EMRs using the Sunrise Clinical Manager platform. Sociodemographic and financial status are captured separately in the Outpatient Administrative System (OAS) which primarily manages patients’ appointments and billing. These clinical and administrative data are transferred from multiple healthcare transactional systems into a single enterprise data repository known as the Electronic Health Intelligence System (eHIntS). These records were extracted by the SHP Research Informatics team for this study using the ETL (Extract, Transform, and Load) database function of the eHIntS. To calculate the cost difference in healthcare expenditure for parents between the old and the new NCIS, we used an estimated average vaccine price from private GP clinics providing fee-for-vaccination services to foreigners and expatriate professionals as a surrogate. The cost data for each vaccine were computed based on the average price cited online from local private GP clinics, which were de-identified as singleton clinics A, B, C and group practices D and E. The mean (SD) cost of vaccination for each child was calculated through the vaccine uptake of PCV13, varicella and influenza in the old and new NCIS, respectively.

The estimated average surrogate vaccine price at private GP clinics was used instead of the SHP vaccine cost charged to the parents, as the SHP vaccine cost is partly subsidized by the nation. Private GP clinics have vaccine prices determined by vendors that are unaffected by subsidies.

### 2.4. Outcomes Definitions

The primary outcomes of this study are the 6- and 12-month childhood vaccination uptake rates. Under the old NCIS, children require 7 doses of vaccines which are administered across 5 visits by 6 months of age. For the new NCIS, children require only 5 doses of vaccines administered in 3 visits by 6 months of age. Children following the old NCIS require the administration of another single dose of vaccine during a visit at 12 months of age. Those who follow the new NCIS regime require another 2 doses of vaccines to be administered at the 12th month. The uptake rate of childhood vaccination may vary due to ethnicity differences and racial disparities [16,17].

### 2.5. Statistical Analysis

All demographic characteristics such as age groups, gender and race were summarized based on vaccination adherence rate. The continuous and categorical variables were described as mean (standard deviation (SD)) and frequency (percentages), respectively. All tests were two-sided and a *p*-value of < 0.05 was set as statistically significant. The differences between categorical and continuous variables, with respect to vaccine fulfillment, were tested using the Chi-Square test and the two-sample *t*-test, respectively. All statistical analyses were performed by using SPSS^®^ version 25.0 (Chicago, IL, USA).

## 3. Results

For our dataset, 21,215 unique children had their childhood immunization records extracted from SHP EMRs. This included (I) 10,916 (51.5%) children vaccinated based on the old NCIS schedule and (II) 10,299 (48.5%) children vaccinated according to the new NCIS. There was a significant difference (*p* < 0.001) in ethnic composition between the national population and the study population. The ethnic composition of the national population was 74.3, 15.0, 7.5 and 1.6% [18].

From Table 2, it can be seen that the 6-month adherence rate for children vaccinated in the new NCIS increased 10.8%, from 65.9 to 76.7%, *p* < 0.001. The 95% confidence interval for the difference in proportions was 0.201 to 0.231 and 0.351 to 0.379 for the old and new NCIS, respectively. The 12-month vaccination adherence rate was raised by 2.1%, from 58.9 to 61.0%, according to the new NCIS, *p* = 0.179. The 95% confidence interval for the difference in proportions was 0.025 to 0.041 and 0.032 to 0.048 for the old and new NCIS, respectively.

Table 3 reports the vaccine uptake of children aged 12 months or older who were scheduled for the 12-month vaccination regiment. The children completing the NCIS were not statistically significant under the two programs. Their adherence rates were 51.1% and 48.9%, respectively, in the old and new NCIS. The estimated average cost per child to the parents was reduced from SGD 400.19 to SGD 0 for the old and new NCIS, respectively. The estimated average cost per child to the healthcare system was reduced from SGD 757.80 to SGD 696.53 for the old and new NCIS, respectively. A total of 81.6% of the children in the old NCIS required six visits to complete the 12-month vaccination, while 89.8% of children in the new NCIS required only four visits to complete all vaccinations required by the 12th month. The children who did not complete 12-month vaccination under the new NCIS were significantly different than those under the old NCIS, specifically, there were more ethnically Chinese and other children, and fewer Malay children.

Among the 1564 children who failed to complete their 12-month vaccination, 53.3% of them were under the old NCIS, while 46.7% were those in the new NCIS. The estimated average cost per child to the parents was reduced from SGD 174.50 to SGD 0 for the old and new NCIS, respectively. The estimated average cost per child to the healthcare system increased from SGD 450.43 (241.41) to SGD 498.99 (210.18) for the old and new NCIS. For children who did not complete the 12-month vaccination despite eligibility, the majority stopped their visit count at the fifth visit (39.0%), or at lower visit counts for the old NCIS. Moreover, 51.6% stopped at the third visit or less for the new NCIS. Many were missing either a few or all of their vaccination visits, and they did not fulfil the 12-month vaccination criteria.

Table 4 shows the average vaccination cost based on the prices from the GP clinics in this study. The new NCIS has a SGD 0 cost of vaccination, as childhood vaccinations are fully subsidized in the new NCIS. Although they are not recommended in the old NCIS, parents do opt to provide influenza and varicella vaccines for the children.

## 4. Discussion

The results show that the childhood vaccination uptake rate did not drop, despite the introduction of the new NCIS. The percentage of children who completed their vaccination increased or remained constant. The 12-month vaccination adherence rate was raised by 2.1%, from 58.9 to 61.0%, as the required visits reduced from six to four [19]. The reduced number of required visits with the new NCIS is intended to reduce the indirect cost, hesitancy, and the inconvenience to the parents of these children. The new NCIS launch is timely as it frees up the workload of primary care nurses to implement the vaccination of large pools of children. The primary healthcare provider can divert human resources to support urgent clinical services, such as during the surge in COVID-19-infected patients in the community during the pandemic.

The COVID-19 pandemic has caused major disruption to healthcare services worldwide, including Singapore [20]. Non-essential healthcare services were deferred, in part, due to safe distancing measures to curb the spread of disease [21]. While the COVID-19 pandemic has impacted childhood vaccination in Singapore [19], both cohorts in our study were vaccinated in the midst of the pandemic. As such, we do not expect the pandemic to significantly impact the differences in vaccine uptake between the two cohorts.

The incorporation of influenza and varicella vaccines as routine vaccinations into the new NCIS meant increased costs of childhood vaccinations for the healthcare system. Prior to the new NCIS, parents who opted for influenza and varicella vaccines for their children were using Medisave to pay for these vaccines. With the new NCIS, these two vaccines are fully subsidized for Singaporean children. The costs of providing the entire national immunization program, including the cost of vaccines and their administration, are estimated from the online market rates of such vaccination services, which are available on the internet. Based on the cost per vaccine administered for the entire local pediatric population, the healthcare expenditure savings from the implementation of the new NCIS are expected to be immense (Table 4). The new NCIS introduced new combination vaccines, such as the 6-in-1 vaccine, which costs less than the combined cost of the 5-in-1 and hepatitis B vaccines, despite being a combination of both. The cost savings, such as reduced vaccine costs, reduced manpower costs due to lesser staff required for lesser injections, and lowered influenza and varicella incidence rates due to higher population immunity, are to be expected. The estimated potential cost saving for each child at the end of the 12-month vaccination period, from the old to the new NCIS, is SGD 15.09, and with 38,672 newborns in 2020, there is an estimated SGD 583,560.48 cost saving for the healthcare system [22].

Despite a significant improvement in costs, there are still children who did not manage to complete the vaccination. The potential reasons which these children did not manage to complete the vaccination might be due to parent’s hesitation towards vaccination, or a lack of trust in healthcare providers/the government [23].

The cost-effectiveness of these vaccines against vaccine-preventable infectious diseases has been established in various populations [24].

Nonetheless, while the vaccine uptake is comparable to the older NCIS, it has yet to attain full adherence by the entire target population of age-appropriate children. The parental, caregiver and children factors which hinder timely vaccination require further study beyond the results which can be generated by EMRs. Minor variations in vaccine uptake are noted among the major ethnic groups in Singapore (Table 1 and Table 2), which will require exploration using other research methodology.

During the period of investigation, families were encouraged to stay indoors and avoid unnecessary inter-personal interactions. [25] While healthcare access was not limited by any pandemic-related restrictions, the various official public health messages to stay at home could potentially have deterred parents from bringing their children for vaccinations at public healthcare or even private GP clinics [26]. Nonetheless, based on the vaccine uptake data, the COVID-19 pandemic did not appear to significantly affect the vaccine uptake despite its disruption to other healthcare services.

Vaccine hesitancy is regarded as a major barrier to preventive healthcare by the World Health Organization. Any changes to the national immunization programme potentially have impacts on vaccine uptake if the healthcare system fails to clarify the intent or address potential concerns about the changes to the vaccine schedule. The positive results of this study show a successful and seamless transition towards a new immunization programme.

### Strengths and Limitations

Being the first community-based study after the launch of NCIS in Singapore, this study provides early insight into the impact of its implementation amidst a pandemic. The evaluation framework in this study can be used by public health researchers to conduct similar assessments of immunization programs in their respective communities.

The data present an opportunity to identify any significant lapse in which remedial measures could be introduced promptly. The NCIS should be regularly reviewed and assessed to further improve the vaccine uptake rate among the target population vis-à-vis healthcare resource allocation for its implementation. An area for potential research is the development of an artificial-intelligence-driven vaccine-predictive model in SHPs to allocate an appropriate level of resources to optimize the implementation of the national childhood immunization program across polyclinics.

We recognize several limitations in our study. The SHP attends to the healthcare needs of approximately one-third of the national population in Singapore and is one of the many healthcare providers which parents can select to administer vaccinations for their children [27]. Information on the vaccine uptake of the different polyclinic clusters in Singapore is not available in this study. This is a potential bias, as we were unable to obtain the vaccination records of all the children in Singapore, and only obtained information from our public healthcare cluster. Being one of the three national public healthcare providers, the findings can be extrapolated to the national population, as approximately one-third of the children who complete their vaccinations will be bound to visit a SHP. Local children who were vaccinated by other providers are recorded in the National Immunization Registry [28], and thus would have been excluded from the SHP data in this study, contributing to the underestimation of the vaccine uptake [29]. Nonetheless, the number of such children is expected to be small, as the polyclinic network is accessible to parents and offers free vaccination services for children. In addition, these children might have had early engagement with polyclinics for neonatal jaundice during the neonatal period, during which childhood vaccination is scheduled for them.

## 5. Conclusions

The new NCIS has resulted in a higher vaccine uptake in children aged between 6 and 12 months. This is desirable because more children are now protected against vaccine-preventable diseases, which can decrease the utilization of the healthcare system as well as morbidity and mortality down the line. There are immediate cost savings to the healthcare system as well, with a reduced number of visits and with lesser nurses required for vaccination.

## Figures and Tables

**Table 1 vaccines-10-01148-t001:** Comparison of the “old” and “new” childhood immunization schedule.

Old National Childhood Immunization Schedule					New National Childhood Immunization Schedule						
Month	HepB	5-in-1 ^1^	PCV13	MMR	Month	6-in-1 ^2^	5-in-1 ^1^	PCV13	MMR	Varicella	Influenza
1st	D1										
					2nd	D1					
3rd		D1	D1								
4th		D2			4th		D1	D1			
5th		D3	D2								
6th	D2				6th	D2		D2			D1
12th				D1	12th				D1	D1	

D1 and D2 refers to dose 1 and dose 2 respectively; ^1^ 5-in-1 refers to a combined Diphtheria, Tetanus, Pertussis, Poliovirus, Haemophilus B vaccine; ^2^ 6-in-1 refers to a combined Diphtheria, Tetanus, Pertussis, Poliovirus, Haemophilus b and Hepatitis B vaccine.

**Table 2 vaccines-10-01148-t002:** Demographics of children in both datasets.

NCIS	Total, *n* (%)	Old, *n* (%)	New, *n* (%)	*p*-Value
	21,213	10,915	10,298	
**Age Group in Months**				0.338
1–5	6789 (32)	3525 (32.3)	3264 (31.7)	
6–11	10,519 (49.6)	5359 (49.1)	5160 (50.1)	
12–13	3905 (18.4)	2031 (18.6)	1874 (18.2)	
**Gender**				0.412
Female	10,359 (48.8)	5360 (49.1)	4999 (48.5)	
Male	10,854 (51.2)	5555 (50.9)	5299 (51.5)	
**Race**				<0.001
Chinese	12,871 (60.7)	6733 (61.7)	6138 (59.6)	
Malay	5390 (25.4)	2780 (25.5)	2610 (25.3)	
Indian	1694 (8.0)	808 (7.4)	886 (8.6)	
Others	1258 (5.9)	594 (5.4)	664 (6.4)	
**6-Month Vaccination ***				<0.001
Did not adhere to schedule	4155 (28.8)	2517 (34.1)	1638 (23.3)	
Adherence to schedule	10,269 (71.2)	4873 (65.9)	5396 (76.7)	
Less than 6 months	6789	3525	3264	
**12-Month Vaccination ***				0.179
Did not adhere to schedule	1564 (40.1)	834 (41.1)	730 (39.0)	
Adherence to schedule	2341 (59.9)	1197 (58.9)	1144 (61.0)	
Less than 12 months	17308	8884	8424	

* 6/12-month vaccination refers to the fulfilment of the cumulative amount of childhood vaccination required at respective months listed above.

**Table 3 vaccines-10-01148-t003:** Demographics of children who fulfilled and did not fulfill 12-month vaccination uptake requirements according to the national childhood immunization schedule.

	Demographics of Children Who Fulfilled 12-Month Vaccination				Demographics of Children Who Did Not Fulfill 12-Month Vaccination Despite Eligibility (12 Months Old)			
	Total, *n* (%)	Old, *n* (%)	New, *n* (%)	*p*-Value	Total, *n* (%)	Old, *n* (%)	New, *n* (%)	*p*-Value
** *n* **	2341	1197 (51.1)	1144 (48.9)		1564	834 (53.3)	730 (46.7)	
**Gender**				0.511				0.735
Female	1197 (51.1)	620 (51.8)	577 (50.4)		762 (48.7)	403 (48.3)	359 (49.2)	
Male	1144 (48.9)	577 (48.2)	567 (49.6)		802 (51.3)	431 (51.7)	371 (50.8)	
**Race**				0.145				0.003
Chinese	1554 (66.4)	806 (67.3)	748 (65.4)		883 (56.5)	450 (54.0)	433 (59.3)	
Malay	562 (24.0)	281 (23.5)	281 (24.6)		166 (10.6)	247 (29.6)	161 (22.1)	
Indian	140 (6.0)	76 (6.3)	64 (5.6)		408 (26.1)	89 (10.7)	77 (10.5)	
Others	85 (3.6)	34 (2.8)	51 (4.5)		107 (6.8)	48 (5.8)	59 (8.1)	
Cost to healthcare system mean (SD)	811.55 (34.20)	837.81 (25.23)	784.06 (15.78)	<0.001	495.35 (225.18)	478.54 (242.78)	514.56 (201.33)	<0.001
Cost to parents mean (SD)		400.19 (29.4)	0			174.50 (143.7)	0	

**Table 4 vaccines-10-01148-t004:** Calculation of vaccine cost and cost distribution for both groups of NCIS children.

Cost of Vaccination at Private Clinics/Clinic Groups	PCV13	MMR	5-in-1	6-in-1	Hepatitis B	Varicella	Influenza
Clinic A	SGD 200.05	SGD 37.45	SGD 107	SGD 128.40	SGD 55.64	SGD 89.88	SGD 48.15
Clinic B *	SGD 125–164	SGD 38	SGD 91	SGD 110–113	SGD 28–65	SGD 75–90	SGD 70
Clinic C	SGD 125	-	-	-	SGD 75	SGD 90	-
GP Group Practice D	SGD 150	-	-	-	-	-	SGD 42.80
GP Group Practice E	SGD 166	-	SGD 86	SGD 133	-	-	SGD 80.25
Mean	SGD 157.20	SGD 37.73	SGD 94.67	SGD 124.30	SGD 59.05	SGD 87.46	SGD 60.30
Cost	Old National Childhood Immunization Schedule			New National Childhood Immunization Schedule			
To healthcare system, mean (SD)	501.31 (257.34)			486.22 (236.52)			
To parents, mean (SD)	194.64 (149.12)			0			

“-” indicates that the clinic/group practice does not offer these vaccines. * Clinic B vaccine prices are a range of estimated prices provided by the clinic subjected to availability. An average of the lowest and highest prices is taken to calculate the mean price of each vaccine.

## Data Availability

Data available on request due to restrictions.

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
