# Peer review of "The Implementation and Impact of a Revised National Childhood Immunization Schedule in an Urban Asian Community"

_vaccines, 2022, doi:10.3390/vaccines10071148_

Round 1
Reviewer 1 Report
This study requires extensive changes to clarify the scientific soundness of this manuscript:
1. The study aim should be clearly defined. The current aim This study aims to the old and new National Childhood Immunization Schedule(NCIS) in Singapore, looking at the number of visits, vaccine uptake, and the cost of vaccinations per child" is unclear both due to the linguistic and unclear spelling as well as methodical issues.
2. The Authors should clearly define the type of the paper "original/article" or "review" - this is crucial to assess this manuscript
3. The methods section should be rewritten. Please provide a logical structure of the text with all the necessary information (Especially for international readers). Please clearly define the timeline, data source, and criteria that were used to assess particular programs.
4. This study is difficult to follow. In the beginning, the authors mentioned "comparison of immunization programs" and then mention the minorities in Singapore (see line 92)
5. Please provide a clear statement, rather than " former and current NCIS". This is unclear!
6. There is a lack of sufficient explanation of the purpose of this study.
7. This study is of limited interest and does not add to international vaccine research.
Author Response
Reviewer Report 1, Reviewer 1
The authors are grateful to the comments from the reviewers. We have revised the manuscript according to the suggestions. We have enlisted the queries from the reviewers and addressed each of them accordingly.
This study requires extensive changes to clarify the scientific soundness of this manuscript:
- The study aim should be clearly defined. The current aim This study aims to the old and new National Childhood Immunization Schedule(NCIS) in Singapore, looking at the number of visits, vaccine uptake, and the cost of vaccinations per child" is unclear both due to the linguistic and unclear spelling as well as methodical issues.
Thank you for the suggestion. We have revised the manuscript to improve its clarity.
- The Authors should clearly define the type of the paper "original/article" or "review" - this is crucial to assess this manuscript
This paper is an Original/Article.
- The methods section should be rewritten. Please provide a logical structure of the text with all the necessary information (Especially for international readers). Please clearly define the timeline, data source, and criteria that were used to assess particular programs.
Thank you for the suggestion. The method has been revised according to the recommendations.
- This study is difficult to follow. In the beginning, the authors mentioned "comparison of immunization programs" and then mention the minorities in Singapore (see line 92)
Thank you for the suggestion. The focus of the paper will be on the “comparison of immunization programs”.
- Please provide a clear statement, rather than " former and current NCIS". This is unclear!
We appreciate the suggestion and have standardized the term to “old” and “new” NCIS.
- There is a lack of sufficient explanation of the purpose of this study.
We attempt to substantiate the background of this study. For example, the use of combination vaccines has been shown to improve the uptake of vaccinations. With the new national childhood schedule, two combination vaccines, the 6-in-1 vaccine and MMRV are made routine. This study is intended to evaluate the impact of the latest childhood immunization schedule with inclusion of new initiatives in the programme.
- This study is of limited interest and does not add to international vaccine research.
We disagree that the study has limited interest. Vaccine hesitancy has been regarded as a major barrier towards preventive health by the World Health Organization. Any change in national immunization programme potentially has impact on vaccine uptake if the healthcare provider fails to clarify the intent or address the concern regarding novel vaccines to be implemented. The results of this study show a seamless transition towards a new NCIS is feasible and can be a model to share with international readers.
Reviewer 2 Report
This manuscript describes a comparison of vaccine uptake before and after implementation of a new National Childhood Immunization regimen, which happened to occur during the COVID-19 pandemic. The authors make important points about how using combination vaccines can facilitate completing vaccinations within fewer visits, which may decrease overall costs and result in increased adherence to the schedule. They also added vaccines to the program that were previously unsupported. This is important information to add to the cannon of vaccine literature.
This manuscript does not seem consistent with a data descriptor as currently written. The title seems overstated-- it's hard to say what of the noted changes are related to the COVID-19 pandemic since other changes were made at the same time. If anything, the pandemic seems to be more of a confounder, which may have led to differential access for different ethnic groups.
Abstract
1. Regarding the phrase “difference in healthcare expenditure to caregivers,” this is unclear. Are we referring to out of pockets costs the caregivers have to pay or government expenditures? Please clarify.
Introduction
1. Recommend clarifying which vaccines are included in the combined vaccines as different abbreviations are used in different areas of the world.
2. Recommend including a reference for the statement that varicella and influenza are growing concerns in Singapore with increasingly prevalent hospitalizations if possible.
3. In the third paragraph, second to last sentence, the authors mention introducing combination vaccines to decrease the required number of visits, but they had previously mentioned using MMR and 5in1 vaccines with the old NCIS. Please clarify which combination vaccines were newly introduced with the new NCIS.
4. One page 2, line 53, the authors state that the impact of the COVID-19 pandemic on vaccination uptake has not been well evaluated but the reference attached is such an evaluation. Please identify the knowledge gap that remains more clearly on this topic.
5. For the paragraph beginning on page 2, line 55: the authors mention that vaccination cost under the former NCIS was only claimable under Medisave. Separately, they mention that recommended vaccines in NCIS are subsidized and offered free at some clinics. It seems from the discussion that these comments only apply to the varicella and influenza vaccines because they were not recommended under the old NCIS? Please clarify this discussion point. Did the payment structure change with the new NCIS for all vaccines or just the varicella and influenza vaccines? Please clarify. I would recommend moving the sentence beginning line 66 to the methods section regarding how cost data were estimated.
6. Regarding the last sentence of the introduction, it seems difficult to infer from these data the effect of the pandemic on vaccine uptake, since 2 changes happened around the same time. It seems data comparing the old NCIS before and after the beginning of the pandemic would provide insight regarding the impact of the pandemic on vaccine uptake, which it outside of the purview of this paper. A potentially more appropriate statement perhaps to be made in the limitations would be COVID-19 as a confounder of the changes seen with the implementation of the new NCIS.
Methods
1. It would be helpful to describe what methods were used to disseminate information/education about the new schedule.
2. Table 1—please define in this table which vaccines are included in 6in1 and 5in1. Is influenza included? If so, please include it in the table
3. On page 4, lines 133-134 the use of t-test and Chi-square tests appear to have been switched around.
4. Please add discussion regarding the methods for secondary aims to the methods section.
Results
1. On page 4 line 137, please clarify whether 21215 refers to individual records or children. Continuing in this paragraph—are these children who were actually vaccinated according to the new or old NCIS or just children that were vaccinated after the new NCIS was implemented? Based on the table, it seems it would be the latter. Please clarify.
2. Please clarify whether the representation of ethnic groups is reflective of the larger population.
3. For Table 2: It is unclear to me what age was used to determine the age group in months. The age at the end of the 12-month study period? Please clarify. Also, there is an asterisk at the bottom of the table but no referring asterisk.
4. For the paragraph beginning on page 4 line 146, do you have confidence intervals for the differences in adherence rates?
5. For Table 3: I’m not sure what Majority of Visit Count refers to. Most number of visits completed? Not sure this should be in the table, maybe just described in the text.
6. In the paragraph beginning line 152 and the following paragraph, no need to restate what is in the table. However, the numbers don’t match, so please double-check which are the correct numbers. Why is the estimated average cost in the paragraph different from that presented in the table? What accounts for the decreased cost for those who completed the vaccination regimen? (If not known, this could be speculated upon in the discussion.) What accounts for the increased cost in those who did not?
7. For table 4: For the “-“, does this mean data not available or that vaccine wasn’t offered in that clinic? Why are ranges reported for clinic B—did the costs change over time? How was this accounted for in the estimates? The cost of vaccination for each child is the estimated amount actually spent to get the vaccination rates achieved in each of the age groups? Did the authors have estimated costs of influenza vaccination?
Discussion
1. Recommend considering adding to the opening paragraph that the percentage of children who completed vaccination increased or remained constant with fewer required visits. This is a big win.
2. For line 194—can the authors extrapolate what kind of savings they might expect?
Author Response
Reviewer Report 1, Reviewer 2
The authors are grateful to the comments from the reviewers. We have revised the manuscript according to the suggestions. We have also enlisted the queries from the reviewers and indicated our responses to each query below.
This manuscript describes a comparison of vaccine uptake before and after implementation of a new National Childhood Immunization regimen, which happened to occur during the COVID-19 pandemic. The authors make important points about how using combination vaccines can facilitate completing vaccinations within fewer visits, which may decrease overall costs and result in increased adherence to the schedule. They also added vaccines to the program that were previously unsupported. This is important information to add to the cannon of vaccine literature.
This manuscript does not seem consistent with a data descriptor as currently written. The title seems overstated-- it's hard to say what of the noted changes are related to the COVID-19 pandemic since other changes were made at the same time. If anything, the pandemic seems to be more of a confounder, which may have led to differential access for different ethnic groups.
Abstract
- Regarding the phrase “difference in healthcare expenditure to caregivers,” this is unclear. Are we referring to out of pockets costs the caregivers have to pay or government expenditures? Please clarify.
Thank you for the comment. We have decided to remove this sentence from the abstract as it is unclear. We will be considering the cost caregivers have to pay for the vaccinations which are not subsidized.
Introduction
- Recommend clarifying which vaccines are included in the combined vaccines as different abbreviations are used in different areas of the world.
Thank you for the recommendations. Vaccines included in the combined vaccines were included in the manuscript to improve clarity.
- Recommend including a reference for the statement that varicella and influenza are growing concerns in Singapore with increasingly prevalent hospitalizations if possible.
Thank you for the recommendations. References on growing concerns of varicella and influenza has been included as [4],[5],[6].
- In the third paragraph, second to last sentence, the authors mention introducing combination vaccines to decrease the required number of visits, but they had previously mentioned using MMR and 5in1 vaccines with the old NCIS. Please clarify which combination vaccines were newly introduced with the new NCIS.
MMR and 5in1 vaccines was previously included in the old NCIS. MMRV and 6in1 vaccines are newly introduced with the new NCIS.
- One page 2, line 53, the authors state that the impact of the COVID-19 pandemic on vaccination uptake has not been well evaluated but the reference attached is such an evaluation. Please identify the knowledge gap that remains more clearly on this topic.
Thank you for the suggestion. The evaluation of COVID-19 pandemic to the childhood immunization schedule have been shift to the discussion part.
- For the paragraph beginning on page 2, line 55: the authors mention that vaccination cost under the former NCIS was only claimable under Medisave. Separately, they mention that recommended vaccines in NCIS are subsidized and offered free at some clinics. It seems from the discussion that these comments only apply to the varicella and influenza vaccines because they were not recommended under the old NCIS? Please clarify this discussion point. Did the payment structure change with the new NCIS for all vaccines or just the varicella and influenza vaccines? Please clarify. I would recommend moving the sentence beginning line 66 to the methods section regarding how cost data were estimated.
There was a change in the payment structure. Previously in the old NCIS, vaccination was paid for with Medisave and other methods. Vaccines in the new NCIS are now fully subsidized by the government and free at public primary care clinics and selected private General Practitioner clinics.
- Regarding the last sentence of the introduction, it seems difficult to infer from these data the effect of the pandemic on vaccine uptake, since 2 changes happened around the same time. It seems data comparing the old NCIS before and after the beginning of the pandemic would provide insight regarding the impact of the pandemic on vaccine uptake, which it outside of the purview of this paper. A potentially more appropriate statement perhaps to be made in the limitations would be COVID-19 as a confounder of the changes seen with the implementation of the new NCIS.
Thank you for the suggestion. Covid-19 will be removed from the sentence and taken as a confounder of the changes seen with the implementation of the new NCIS.
Methods
- It would be helpful to describe what methods were used to disseminate information/education about the new schedule.
The new NCIS was introduced to the public, polyclinics and general practitioner clinics through MOH circular. Roadshows were conducted to introduce the new NCIS and various healthcare website published infographics to increase awareness of the NCIS.
- Table 1—please define in this table which vaccines are included in 6in1 and 5in1. Is influenza included? If so, please include it in the table
Vaccines included in 5in1, 6in1 are included as notes below Table 1. Influenza is included in Table 1.
- On page 4, lines 133-134 the use of t-test and Chi-square tests appear to have been switched around.
Thanks for pointing out. We have made the necessary changes.
- Please add discussion regarding the methods for secondary aims to the methods section.
Discussion regarding methods for secondary aims has been included in the methods section.
Results
- On page 4 line 137, please clarify whether 21215 refers to individual records or children. Continuing in this paragraph—are these children who were actually vaccinated according to the new or old NCIS or just children that were vaccinated after the new NCIS was implemented? Based on the table, it seems it would be the latter. Please clarify.
21215 children’s immunization records were obtained. 10916 (51.5%) children followed the old NCIS and vaccinated accordingly. 10299 (48.5%) children followed the new NCIS and vaccinated accordingly.
- Please clarify whether the representation of ethnic groups is reflective of the larger population.
Representation of ethnic group is not reflective of the large population and statistically significantly different.
Chinese: 74.2%, Malay: 13.7%, Indian: 8.9% and Others: 3.2%
Relevant reference has been included as well.
- For Table 2: It is unclear to me what age was used to determine the age group in months. The age at the end of the 12-month study period? Please clarify. Also, there is an asterisk at the bottom of the table but no referring asterisk.
The age is determined at the end of the 12-month study period, at 29th December 2020 for the old NCIS and at 29th December 2021 for the new NCIS.
- For the paragraph beginning on page 4 line 146, do you have confidence intervals for the differences in adherence rates?
The confidence interval has been included in the results in text.
- For Table 3: I’m not sure what Majority of Visit Count refers to. Most number of visits completed? Not sure this should be in the table, maybe just described in the text.
The majority of visit counts is the highest occurrence of the amount of visit each child made to the polyclinic for vaccination. Information will be removed from the table and described in text.
- In the paragraph beginning line 152 and the following paragraph, no need to restate what is in the table. However, the numbers don’t match, so please double-check which are the correct numbers. Why is the estimated average cost in the paragraph different from that presented in the table? What accounts for the decreased cost for those who completed the vaccination regimen? (If not known, this could be speculated upon in the discussion.) What accounts for the increased cost in those who did not?
Thank you for pointing out the differences. Necessary changes have been made. We have decided to portray the cost difference in another manner. Cost of vaccination are considered as cost to the caregivers who pay for vaccinations which are not subsidized by the healthcare system.
- For table 4: For the “-“, does this mean data not available or that vaccine wasn’t offered in that clinic? Why are ranges reported for clinic B—did the costs change over time? How was this accounted for in the estimates? The cost of vaccination for each child is the estimated amount actually spent to get the vaccination rates achieved in each of the age groups? Did the authors have estimated costs of influenza vaccination?
The vaccines were not offered in the clinic. The range is an estimate given by clinic B for the price of their vaccines. These vaccine prices are subjected to availability. An average of the highest and lowest vaccine price is taken. Yes, the cost of vaccination per child is the estimated amount actually spent to get the vaccination rates achieved in each of the age groups. Estimated costs of influenza vaccination has been included.
Discussion
- Recommend considering adding to the opening paragraph that the percentage of children who completed vaccination increased or remained constant with fewer required visits. This is a big win.
Thank you for the suggestion. We have incorporated the positive results into the first paragraph of the discussion.
- For line 194—can the authors extrapolate what kind of savings they might expect?
Cost savings such as reduced vaccines cost, reduced manpower costs and lowered influenza and varicella incidence costs are to be expected.
Reviewer 3 Report
In this report, Tan et al compare the uptake of various childhood vaccines and the frequency of primary healthcare provider visits for vaccination over the course of one year before and after the launch of a newly revised National Childhood Immunization Schedule (NCIS) in Singapore. Overall, the manuscript is logically organized and written clearly. The study design is scientifically sound, and the electronic medical records data extracted from the SingHealth Polyclinics (SHP) thoughtfully selective. Though the study only revealed marginally higher vaccine uptake in young children (6-12 mo), its methodology and data strongly support that the NCIS be subject to regular reviews and EMT data critically analyzed to identify potential hurdles to vaccine uptake, aid in the allocation of resources to best support a national childhood immunization program and provide a greater understanding of healthcare expenditure savings realized national childhood immunization program. My minor comments are:
1. Line 12: unnecessary hyphen; replace with the full word “vaccinations”
2. Line 40: add period after the word “vaccine”
3. Line 42: though MMR is defined in line 34, the abbreviation MMRV needs to be fully defined
4. Line 47: capitalize the c in “covid-19”
5. Line 61: the citations [13] [14] appear before the period. This is different from how the other citations are formatted. I encourage consistency and use of the publisher’s preferred format
6. Line 73: unnecessary hyphen; replace with the full word “various”
7. Line 94: unnecessary hyphen; replace with the full word “October”
8. Line 129: unnecessary hyphen; replace with the full word “summarized”
9. Line 129: unnecessary hyphen; replace with the full word “described”
10. Line 154: unnecessary hyphen; replace with the full word “Indian”
11. Line 161: unnecessary hyphen; replace with the full word “included”
12. Line 162: unnecessary hyphen; replace with the full word “Chinese”
13. Line 187: unnecessary hyphen; replace with the full word “Medisave”
14. Line 195: unnecessary hyphen; replace with the full word “infection”
15. Line 196: unnecessary hyphen; replace with the full word “significant”
16. Line 200: unnecessary hyphen; replace with the full word “parental”
17. Line 213: unnecessary hyphen; replace with the full word “provides”
18. Line 215: unnecessary hyphen; replace with the full word “researchers”
19. Line 219: unnecessary hyphen; replace with the full word “assessed”
20. Line 222: unnecessary hyphen; replace with the full word “vaccine”
21. Line 226: unnecessary hyphen; replace with the full word “vaccinated”
Author Response
Reviewer Report 1, Reviewer 3
The authors are grateful to the comments from the reviewers. We have revised the manuscript according to the suggestions. We have also enlisted the queries from the reviewers and indicated our responses to each query below.
- Line 12: unnecessary hyphen; replace with the full word “vaccinations”
Hyphen has been removed.
- Line 40: add period after the word “vaccine”
Hyphen has been removed.
- Line 42: though MMR is defined in line 34, the abbreviation MMRV needs to be fully defined
Measles, Mumps, Rubella and Varicella (MMRV) has been defined in the writing.
- Line 47: capitalize the c in “covid-19”
Covid-19 has been capitalized in the writing.
- Line 61: the citations [13] [14] appear before the period. This is different from how the other citations are formatted. I encourage consistency and use of the publisher’s preferred format
Citations [13] [14] has been moved to appear after the period.
- Line 73: unnecessary hyphen; replace with the full word “various”
Hyphen has been removed.
- Line 94: unnecessary hyphen; replace with the full word “October”
Hyphen has been removed.
- Line 129: unnecessary hyphen; replace with the full word “summarized”
Hyphen has been removed.
- Line 129: unnecessary hyphen; replace with the full word “described”
Hyphen has been removed.
- Line 154: unnecessary hyphen; replace with the full word “Indian”
Hyphen has been removed.
- Line 161: unnecessary hyphen; replace with the full word “included”
Hyphen has been removed.
- Line 162: unnecessary hyphen; replace with the full word “Chinese”
Hyphen has been removed.
- Line 187: unnecessary hyphen; replace with the full word “Medisave”
Hyphen has been removed.
- Line 195: unnecessary hyphen; replace with the full word “infection”
Hyphen has been removed.
- Line 196: unnecessary hyphen; replace with the full word “significant”
Hyphen has been removed.
- Line 200: unnecessary hyphen; replace with the full word “parental”
Hyphen has been removed.
- Line 213: unnecessary hyphen; replace with the full word “provides”
Hyphen has been removed.
- Line 215: unnecessary hyphen; replace with the full word “researchers”
Hyphen has been removed.
- Line 219: unnecessary hyphen; replace with the full word “assessed”
Hyphen has been removed.
- Line 222: unnecessary hyphen; replace with the full word “vaccine”
Hyphen has been removed.
- Line 226: unnecessary hyphen; replace with the full word “vaccinated”
Hyphen has been removed.
Round 2
Reviewer 1 Report
The manuscript was significantly revised.
However, some changes are still needed:
1. Sampling method: please consider how the sampling methods affect the results? Is the study representative? Can we extrapolate the findings to the whole population or some sub-populations?
2. How the location of the SingHealth Polyclinics may affect the results? E.g. facilities located in the business district may differ from the poor districts
3. Please strengthen the rationale for this study. Why this study is important and why this method may be useful? There is some inconsistency between the study aim and the presented findings.
Author Response
The manuscript was significantly revised.
However, some changes are still needed:
- Sampling method: please consider how the sampling methods affect the results? Is the study representative? Can we extrapolate the findings to the whole population or some sub-populations?
This is a cluster sampling, where vaccine records of children from our polyclinics cluster located in the east are taken. The findings can be extrapolated to the whole population as SingHealth Polyclinics is one of the three major public healthcare providers for Singapore where the vaccine records are approximately one third of the national population.
- How the location of the SingHealth Polyclinics may affect the results? E.g. facilities located in the business district may differ from the poor districts.
The location of SHP may affect the results but as we do not have information on the vaccine uptake of the different polyclinic cluster, we cannot describe the extent of the effects on the results.
- Please strengthen the rationale for this study. Why this study is important and why this method may be useful? There is some inconsistency between the study aim and the presented findings.
This study is important as vaccine hesitancy has been regarded as a major barrier towards preventive health by the World Health Organization. Any changes to the national immunization programme potentially has impact on vaccine uptake if the healthcare system fails to clarify the intent or address potential concern on the changes to the vaccine schedule. The results of the study show a successful seamless transition towards a new national immunization schedule and a model which can be share with international readers.
Reviewer 2 Report
The revised manuscript is much more clear. I have a few minor suggestions at this point:
Introduction
1. In the sentence beginning line 26 describing the vaccines, the use of semi-colons between the vaccines would make the meaning of the sentence more clear, similar to what is used in a similar sentence later on.
2. In the last paragraph of the introduction, COVID-19 is mentioned, I would recommend not mentioning it as an aim of the study because there is no methodology presented to assess the effect of COVID-19 as a confounder. The discussion in the discussion section is adequate.
Table 2
1. I would spell out the meaning of NCIS in the title and include the abbreviation in parentheses next to it (NCIS), since the abbreviation is used in the table.
Materials and Methods
1. The first part of the section 2.4 on page 3 starting line 104 is redundant with section 2.2 and should be removed.
2. In section 2.5 on page 3, the sentence starting line 123, I would remove the word all in the sentence, as it suggests that all vaccines in the NCIS can be completed by 6 months of age, which conflicts with Table 2 and isn't true. Something like, "Under the old NCIS, 7 doses were required by 6 months of age and were administered over 5 visits" with something similar for the sentence starting line 124 excluding the word all.
Results
1. Again, I would recommend removing the delineation of ethnic composition of the two groups from the text, since it is clearly presented in the table a few centimeters away.
2. I would make more emphasis of the fact that the differences of both groups differ from that of the general population. The reason for this would be interesting to speculate upon in the discussion. (Differing acceptance of or access to vaccination by ethnic group?)
3. The sentence beginning line 146 about how children's ages are calculated should be moved to the methods section 2.1.
4. Thank you for including the confidence intervals; however, it is unclear what these confidence intervals represent. They look like a ratio. Since the percentage difference is what is appropriately presented, a confidence interval for that percent difference would be more meaningful here.
5. The adherence rates mentioned on page 5 line 159-160 do not match those presented in Table 3. The percentages presented in this sentence add up to 100%.
6. Again, I would not restate the ethnicity breakdown in the text. To me the take home is that the children who did not complete 12 month vaccination under the new NCIS are significantly different ethnically than under the old NCIS, specifically more ethnically Chinese and other children and fewer Malay children-- I would reiterate that point in the sentence beginning line 169. For the sentence starting at line 160, I would simply summarize that the children completing the NCIS were not statistically significantly different under the two programs.
7. For the sentence starting on line 165, instead of the 12th month vaccination, I would use the phrase "all vaccinations required by 12 months". Otherwise, it sounds like you are only referring to the vaccinations scheduled to be received at the 12 month visit.
8. I'm not sure why the sentence beginning on line 168 is helpful to your case.
9. It's interesting that for those children who did not complete vaccination despite the significant improvement in the cost. It would be helpful to suggest in the discussion the need to explore the barriers to completing vaccination for those children, since cost is no longer the issue.
10. For the sentence beginning line 174, it might be more clear to reword, "For children who did not complete the 12th month vaccination despite eligibility, most (39%) completed through the 5th visit under the old NCIS and through the 3rd visit (51.6%) under the new NCIS." This makes the point that if you can reduce the barriers to getting that 4th visit in under the new NCIS, you only have 48.4% to get to full vaccination rates compared with 61% under the old system, a point which could be made in the discussion.
11. On page 6, beginning line 182, I would move this information to the methods. I would state here that costs under the new NCIS are reduced both for the healthcare system and for parents out of pocket per child.
Table 4
1. I don't understand the utility of comparing the children who completed 12 month vaccines by 12 months or by 13 months by whether they did or did not complete the series. Perhaps this point would be better made in the text and not in the first row of this table.
Table 5.
1. Please define NCIS so the table can stand alone
Discussion
1. In the sentence starting line 204, I would clarify for whom the increased costs are incurred (I'm assuming the healthcare system).
2. Line 207 is the first time you mention Baby Bonus accounts. I would leave this out since you have not previously defined it (previously referring to other means).
3. One could estimate potential cost savings by multiplying the difference in costs per child and the number of children that are vaccinated each year, which makes quite a convincing argument.
4. In the sentence beginning line 247, this is your first mention of underestimation of vaccine uptake using your methods. Are the children who are vaccinated in other systems likely to be demographically different? Could this be a source of potential bias? Are there other providers for neonatal jaundice who also provide vaccines?
Conclusion
1. I would recommend making two separate points: the new NCIS has resulted in higher vaccine uptake-- this is desirable because more children are protected against more preventable diseases which can decrease utilization of the healthcare system for treating those diseases as well as morbidity and mortality down the line. The second point is that there are more immediate savings to the healthcare system with a reduced number of visits.
Author Response
The revised manuscript is much more clear. I have a few minor suggestions at this point:
Introduction
- In the sentence beginning line 26 describing the vaccines, the use of semi-colons between the vaccines would make the meaning of the sentence clearer, similar to what is used in a similar sentence later on.
Thank you for the advice. We have used semi-colons to improve the clarity of the sentence.
- In the last paragraph of the introduction, COVID-19 is mentioned, I would recommend not mentioning it as an aim of the study because there is no methodology presented to assess the effect of COVID-19 as a confounder. The discussion in the discussion section is adequate.
COVID-19 has been removed from the introduction.
Table 2
- I would spell out the meaning of NCIS in the title and include the abbreviation in parentheses next to it (NCIS), since the abbreviation is used in the table.
Thank you for the advice. We have edited the table accordingly.
Materials and Methods
- The first part of the section 2.4 on page 3 starting line 104 is redundant with section 2.2 and should be removed.
Thank you for the advice. It has been removed.
- In section 2.5 on page 3, the sentence starting line 123, I would remove the word all in the sentence, as it suggests that all vaccines in the NCIS can be completed by 6 months of age, which conflicts with Table 2 and isn't true. Something like, "Under the old NCIS, 7 doses were required by 6 months of age and were administered over 5 visits" with something similar for the sentence starting line 124 excluding the word all.
Thank you for the advice. We have rephrased and improved the clarity of the sentence.
Results
- Again, I would recommend removing the delineation of ethnic composition of the two groups from the text, since it is clearly presented in the table a few centimeters away.
Thank you for the advice. We have removed the ethnic composition from the text.
- I would make more emphasis of the fact that the differences of both groups differ from that of the general population. The reason for this would be interesting to speculate upon in the discussion. (Differing acceptance of or access to vaccination by ethnic group?)
Thank you for the advice. There is emphasis placed on the differences of both group in the discussion.
- The sentence beginning line 146 about how children's ages are calculated should be moved to the methods section 2.1.
Thank you for the advice. The information has been shifted to the methods section.
- Thank you for including the confidence intervals; however, it is unclear what these confidence intervals represent. They look like a ratio. Since the percentage difference is what is appropriately presented, a confidence interval for that percent difference would be more meaningful here.
Thank you for the advice. The confidence interval for the percentage difference has been included.
- The adherence rates mentioned on page 5 line 159-160 do not match those presented in Table 3. The percentages presented in this sentence add up to 100%.
The adherence rate in line 159-160 is different from Table 3 as adherence rate in line 159-160 describes children aged 12-13 months old only.
- Again, I would not restate the ethnicity breakdown in the text. To me the take home is that the children who did not complete 12 month vaccination under the new NCIS are significantly different ethnically than under the old NCIS, specifically more ethnically Chinese and other children and fewer Malay children-- I would reiterate that point in the sentence beginning line 169. For the sentence starting at line 160, I would simply summarize that the children completing the NCIS were not statistically significantly different under the two programs.
Thank you for the advice. These information has been included accordingly for the two programs.
- For the sentence starting on line 165, instead of the 12th month vaccination, I would use the phrase "all vaccinations required by 12 months". Otherwise, it sounds like you are only referring to the vaccinations scheduled to be received at the 12 month visit.
Thank you for the advice. We have rephrased the sentence to improve clarity.
- I'm not sure why the sentence beginning on line 168 is helpful to your case.
Thank you for the advice. This explains the demographics of children who failed the 12-month vaccination criteria.
- It's interesting that for those children who did not complete vaccination despite the significant improvement in the cost. It would be helpful to suggest in the discussion the need to explore the barriers to completing vaccination for those children, since cost is no longer the issue.
Thank you for the advice. Discussion on the possible reasons which children has barriers to vaccination has been included, namely vaccine hesitancy.
- For the sentence beginning line 174, it might be more clear to reword, "For children who did not complete the 12th month vaccination despite eligibility, most (39%) completed through the 5th visit under the old NCIS and through the 3rd visit (51.6%) under the new NCIS." This makes the point that if you can reduce the barriers to getting that 4th visit in under the new NCIS, you only have 48.4% to get to full vaccination rates compared with 61% under the old system, a point which could be made in the discussion.
Thank you for the advice. This has been rephrased to improve clarity.
- On page 6, beginning line 182, I would move this information to the methods. I would state here that costs under the new NCIS are reduced both for the healthcare system and for parents out of pocket per child.
Thank you for the advice. This information has been moved to the methods section.
Table 4
- I don't understand the utility of comparing the children who completed 12 month vaccines by 12 months or by 13 months by whether they did or did not complete the series. Perhaps this point would be better made in the text and not in the first row of this table.
Thank you for the advice. We have changed it so that table 4 does not introduce the age of the children as they are aged 12-13 and not significant in comparing the results.
Table 5.
- Please define NCIS so the table can stand alone
NCIS has been defined in the table.
Discussion
- In the sentence starting line 204, I would clarify for whom the increased costs are incurred (I'm assuming the healthcare system).
Thank you for the advice. The costs have been indicated for the healthcare system.
- Line 207 is the first time you mention Baby Bonus accounts. I would leave this out since you have not previously defined it (previously referring to other means).
Thank you for the advice. Baby bonus accounts were removed from the writing.
- One could estimate potential cost savings by multiplying the difference in costs per child and the number of children that are vaccinated each year, which makes quite a convincing argument.
Thank you for the advice. Potential cost savings has been introduced with $15.09 cost saving per child and with 38672 children born in 2021, there is $583,560.48 cost savings.
- In the sentence beginning line 247, this is your first mention of underestimation of vaccine uptake using your methods. Are the children who are vaccinated in other systems likely to be demographically different? Could this be a source of potential bias? Are there other providers for neonatal jaundice who also provide vaccines?
The children in different healthcare cluster is unlikely to be demographically different. For 2021, 38672 children born in Singapore were of the following ethnic composition. Chinese: 56.9%, Malay: 20.3%, Indians: 12.2% and others: 10.6%. This is different from the ethnic composition of the new NCIS. However, we do not have the vaccine uptake of children from other public healthcare groups
Conclusion
- I would recommend making two separate points: the new NCIS has resulted in higher vaccine uptake-- this is desirable because more children are protected against more preventable diseases which can decrease utilization of the healthcare system for treating those diseases as well as morbidity and mortality down the line. The second point is that there are more immediate savings to the healthcare system with a reduced number of visits.
Thank you for the advice. These point has been included in the conclusion.